

# Serum-deprived differentiated neuroblastoma F-11 cells express functional dorsal root ganglion neuron properties

Valentina Pastori[1], Alessia D'Aloia[2], Stefania Blasa[2] and Marzia Lecchi[1]

[1] Department of Biotechnology and Biosciences and Milan Center for Neuroscience, University of Milano-Bicocca, Milan, Italy
[2] Department of Biotechnology and Biosciences, University of Milano-Bicocca, Milan, Italy

Corresponding author
Marzia Lecchi,
marzia.lecchi1@unimib.it

## ABSTRACT

The isolation and culture of dorsal root ganglion (DRG) neurons cause adaptive changes in the expression and regulation of ion channels, with consequences on neuronal excitability. Considering that not all neurons survive the isolation and that DRG neurons are heterogeneous, it is difficult to find the cellular subtype of interest. For this reason, researchers opt for DRG-derived immortal cell lines to investigate endogenous properties. The F-11 cell line is a hybridoma of embryonic rat DRG neurons fused with the mouse neuroblastoma line N18TG2. In the proliferative condition, F-11 cells do not display a gene expression profile correspondent with specific subclasses of sensory neurons, but the most significant differences when compared with DRGs are the reduction of voltage-gated sodium, potassium and calcium channels, and the small amounts of *TRPV1* transcripts. To investigate if functional properties of mature F-11 cells showed more similarities with those of isolated DRG neurons, we differentiated them by serum deprivation. Potassium and sodium currents significantly increased with differentiation, and biophysical properties of tetrodotoxin (TTX)-sensitive currents were similar to those characterized in small DRG neurons. The analysis of the voltage-dependence of calcium currents demonstrated the lack of low threshold activated components. The exclusive expression of high threshold activated $Ca^{2+}$ currents and of TTX-sensitive $Na^+$ currents correlated with the generation of a regular tonic electrical activity, which was recorded in the majority of the cells (80%) and was closely related to the activity of afferent TTX-sensitive A fibers of the proximal urethra and the bladder. Responses to capsaicin and substance P were also recorded in ~20% and ~80% of cells, respectively. The percentage of cells responsive to acetylcholine was consistent with the percentage referred for rat DRG primary neurons and cell electrical activity was modified by activation of non-NMDA receptors as for embryonic DRG neurons. These properties and the algesic profile (responses to pH5 and sensitivity to both ATP and capsaicin), proposed in literature to define a sub-classification of acutely dissociated rat DRG neurons, suggest that differentiated F-11 cells express receptors and ion channels that are also present in sensory neurons.

## INTRODUCTION

The F-11 cell line is a hybrid obtained by fusion of embryonic rat dorsal root ganglion (DRG) and mouse neuroblastoma by *Platika et al. (1985)*. These cells have been widely used in the past years in proliferating conditions to study properties of DRG neurons, but their transcriptomic analysis appeared only 2 years ago by means of Illumina next-generation sequencing, revealing that their gene expression profile did not resemble any specific defined DRG subclass (*Yin, Baillie & Vetter, 2016*).

F-11 cells could also be differentiated into functional neurons by retinoic acid incubation (*Chiesa et al., 1997*; *Ambrosino et al., 2013*), and by their maintenance on biocompatible substrates (neoglucosylated collagen matrices, *Russo et al., 2014*). Although the acquisition by differentiated F-11 cells of characteristic neuronal electrophysiological properties, such as sodium, potassium and calcium currents, and action potential (AP) firing, have been described, their properties as sensory neurons have not been documented so far.

Here, we show for the first time an exhaustive characterization of the electrophysiological properties of F-11 cells differentiated by serum deprivation. Our aim was to investigate if differentiated F-11 cells manifest similarities with DRG neurons in order to verify whether they are an adequate model for studying mechanisms involved in the detection and transmission of noxious stimuli.

## MATERIALS AND METHODS

### Cell cultures

F-11 cells (mouse neuroblastoma N18TG-2 x rat DRG, ECACC Cat#08062601 RRID: CVCL_H605; *Platika et al., 1985*) were seeded at 60,000 cells/35 mm dish and were maintained without splitting in low serum medium for almost 2 weeks to induce differentiation. The complete composition of the medium was: Dulbecco's modified Eagle's medium (Cat#D6546; Sigma-Aldrich, St. Louis, MO, USA), 2 mM glutamine (Sigma-Aldrich, St. Louis, MO, USA), 1% fetal bovine serum (FBS, Cat# F2442; Sigma-Aldrich, St. Louis, MO, USA). The cells were incubated at 37 °C in a humidified atmosphere with 5% $CO_2$, receiving fresh medium twice per week. F-11 cells maintained in 10% FBS medium were used as control. Morphological and functional analysis were performed at 10–14 days of differentiation, whereas cells growing in the non-differentiation medium underwent the same analysis at 7 days to avoid dramatic cell death due to confluence.

### Immunofluorescence

F-11 cells were plated at a density of 60,000 cells on coverslips pre-treated with gelatin from porcine skin (Sigma-Aldrich, St. Louis, MO, USA). Cells were then maintained in DMEM medium supplemented with 1% or 10% serum. After 10 and 7 days, respectively, differentiated and undifferentiated cells were fixed for 10 min in 3.7% paraformaldehyde in

phosphate-buffer saline (PBS), permeabilized for 4 min with 0.1% Triton X-100 in PBS, and stained with monoclonal anti-NeuN/Fox3 produced in mouse primary antibody (1:150, Cat#MAB-94161; Immunological Sciences, Rome, Italy). Incubation with secondary antibody Cyanine3 goat anti-mouse IgG (H+L) (1:200, Cat#A10521, RRID: AB_2534030; Thermo Fisher, Waltham, MA, USA) was maintained for 45 min. After washout in 1× PBS, DAPI (Sigma-Aldrich, St. Louis, MO, USA) was added at the final concentration of 1 μg/ml in 1× PBS and incubation was maintained for 10 min at room temperature. After washing, the slides were mounted and photographed using an A1RNikon (Nikon, Tokyo, Japan) laser scanning fluorescence confocal microscope at 40× magnification. A total of 16–20 z-stack images from 10 different fields for each condition were taken. For this analysis three coverslips of cells maintained in 10% serum and three coverslips of cells in 1% serum were prepared. Transmission images were captured with a Leica TCSSP2 confocal microscope equipped with a 100×/PH3 oil immersion objective. The images were acquired from three cultures of cells maintained in 10% serum and from three cultures of cells in 1% serum.

## Functional analysis by the patch-clamp technique

Functional characterization of the electrophysiological properties of F-11 cells was performed by the patch-clamp technique in the whole-cell configuration. For the experiments, culture media were replaced by a standard extracellular solution which contained (mM): NaCl 135, KCl 2, $CaCl_2$ 2, $MgCl_2$ 2, hepes 10, glucose 5, pH 7.4. The standard pipette solution contained the following (mM): potassium aspartate 130, NaCl 10, $MgCl_2$ 2, $CaCl_2$ 1.3, EGTA 10, hepes 10, pH 7.3. Recordings were acquired by the pClamp8.2 software (pClamp, RRID:SCR_011323) and the MultiClamp 700A amplifier (Axon Instruments; Molecular Devices, LLC., San Jose, CA, USA). Resting membrane potential ($V_{rest}$) and APs were monitored in the current-clamp mode. Cells that did not exhibited spontaneous firing were depolarized with 1 s-long current pulses under conditioning hyperpolarization at −75/−80 mV to verify their capability to generate repetitive spiking. In the voltage-clamp mode, series resistance errors were compensated for a level of up to 85–90%. Sodium ($I_{Na}$) and potassium ($I_K$) currents were recorded by applying a standard protocol: starting from a holding potential of −60 mV, cells were conditioned at −90 mV for 500 ms and successively tested by depolarizing potentials in 10 mV-increments, from −80 to +40 mV. $Na^+$ and $K^+$ currents were isolated by applying 0.3–1 μM tetrodotoxin (TTX) or 10 mM tetraethylammonium (TEA) in the bath. To determine current densities ($I_{Na}$ and $I_K$), the maximal inward and outward current intensities were respectively chosen for $Na^+$ and $K^+$ currents. $Na^+$ channel biophysical properties of activation and inactivation were studied by using a pipette solution containing (mM): CsF 105, CsCl 27, NaCl 5, $MgCl_2$ 2, EGTA 10, hepes 10, pH 7.3. The voltage-dependence of activation was determined by the above mentioned protocol, whereas for the inactivation properties the protocol consisted in a conditioning step with amplitude from −105 to 0 mV and duration of 600 ms, followed by a test at −10 mV.

Ether-à-go-go-related gene (ERG) potassium currents ($I_{erg}$) were recorded by using an extracellular solution containing a higher $K^+$ concentration (40 mM), and by imposing a

standard protocol which, starting from an holding potential of $-60$ mV, conditioned the cell membrane for 15 s from $-80$ to $+20$ mV (20 mV increments) and successively hyperpolarized at $-120$ mV to evoke the tail current. Normalized tail currents were interpolated with a Boltzmann function to obtain the activation curve. To study the voltage-dependence of ERG channel inactivation, a stimulation protocol was applied which, starting from a holding potential of 0 mV, applied voltage steps with duration of ~200 ms and amplitude from $+20$ mV to $-140$ mV. ERG potassium current was analyzed by using WAY-123,398 (1 µM) as selective blocker and CsCl (5 mM) as further inhibitor.

$Ba^{2+}$ currents through voltage-dependent $Ca^{2+}$ channels were recorded under conditions which suppressed $Na^+$ and $K^+$ currents: the culture medium was replaced by an external solution composed by TEA-chloride 130 mM, $BaCl_2$ 10 mM, $MgCl_2$ 1 mM, hepes 10 mM, TTX 1 µM, glucose 10 mM, pH 7.3; the internal pipette solution contained CsCl 140 mM, $MgCl_2$ 4 mM, hepes 10 mM, EGTA 10 mM, Na-ATP 2 mM, pH 7.2. To examine the current-voltage relationship (IV) of $I_{Ba}$, the cells were depolarized with increasing 10 mV steps from $-60$ to $+50$ mV. Moreover, to analyze the contribution of high threshold ($I_{Ba(high)}$) and low threshold ($I_{Ba(low)}$) currents, cells were stimulated by two different protocols: $I_{Ba(high)}$ were elicited by depolarizing pulses to 0 mV for 150 ms from an holding potential of $-80$ mV, whereas for $I_{Ba(low)}$ the holding potential was imposed at $-90$ mV and the test potential was clamped at $-50$ mV. Addition of 200 µM $CdCl_2$ (Sigma-Aldrich, St. Louis, MO, USA) in the extracellular solution confirmed that $I_{Ba}$ flowed through voltage-dependent calcium channels, and nifedipine (5 µM, Sigma-Aldrich, St. Louis, MO, USA) was used to isolate the component of $I_{Ba}$ through L-type voltage-dependent calcium channels.

Acetylcholine (ACh) currents were evoked by applying ACh (1 mM, Sigma-Aldrich, St. Louis, MO, USA) and were inhibited by the nicotinic ACh receptor antagonist d-tubocurarine (DTC, 1 µM, Sigma-Aldrich, St. Louis, MO, USA). Glutamate (1 mM, Sigma-Aldrich, St. Louis, MO, USA), CNQX (6-cyano-7-nitroquinoxaline-2,3-dione, 10 µM, Tocris, Bristol, UK) and AP5 ((2R)-amino-5-phosphonovaleric acid, 40 µM, Tocris, Bristol, UK) were bath applied to evaluate the functional expression of α-amino-3-hydroxy-5-methyl-4-isoxazolepropionic acid (AMPA) and N-methyl-D-aspartic acid (NMDA) receptors, respectively. To allow NMDA receptor activation, $Mg^{2+}$-free extracellular solution was used.

To study acidic condition responses mediated by transient receptor potential vanilloid 1 (TRPV1) non-selective cation channel and acid-sensing ion channels (ASIC), cells were superfused with a solution containing (mM): NaCl 135, KCl 2, $CaCl_2$ 2, $MgCl_2$ 2, MES 10, glucose 5, pH 5, or pH 6. To investigate the expression of proteins involved in nociception, capsaicin (CAPS, 3 µM, Sigma-Aldrich, St. Louis, MO, USA) and substance P (SP, 2 µM, Sigma-Aldrich, St. Louis, MO, USA) were applied.

### Data analysis
Patch-clamp experiments were performed on a minimum of two and on a maximum of twenty independent cultures for each condition. For the analysis, Origin 8

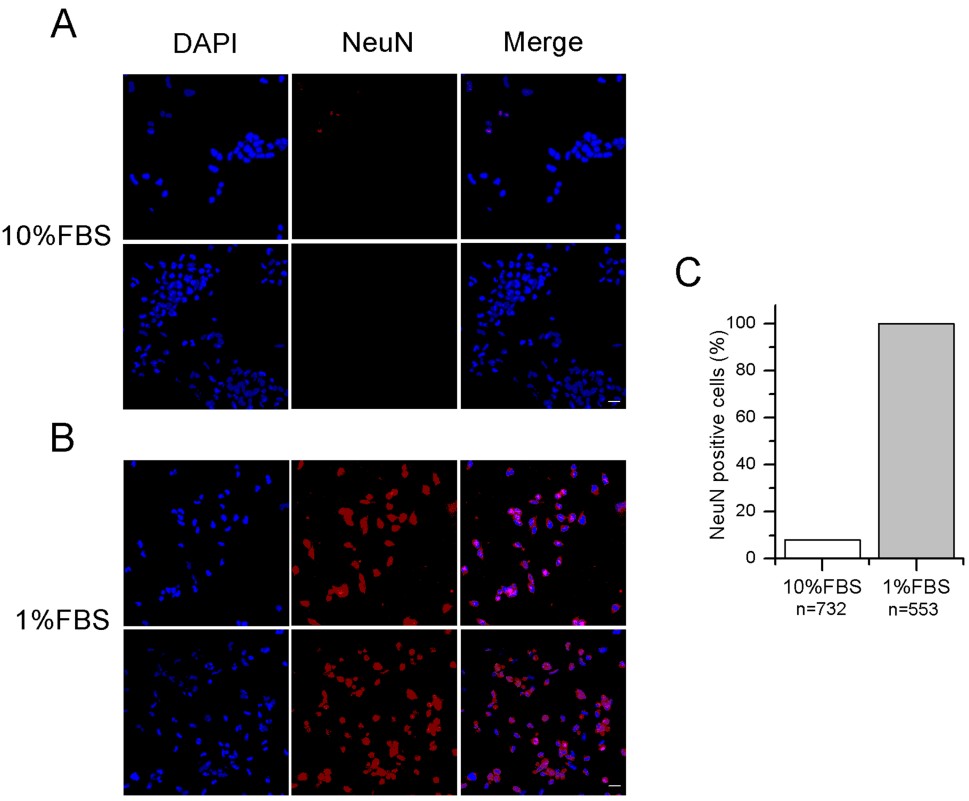

**Figure 1 Differentiated F-11 cells express the neuronal nuclear antigen NeuN.** (A, B) The panels illustrate NeuN staining in red, DAPI in blue and the color overlay (merged) in F-11 cells maintained in 10% FBS and 1% FBS, respectively. A total of 16–20 z-stack images from for each condition were taken. (C) Quantification of NeuN positive cells (histograms) in 10 different fields confirmed no or minor expression of this nuclear marker in 10% FBS compared to 1% FBS cultures. Fluorescence images were captured with a laser scanning fluorescence confocal microscope at 40× magnification. Scale bar, 20 μm.                                                               

(RRID:SCR_014212; Microcal Inc., Northampton, MA, USA) and Excel (Microsoft, Redmond, WA, USA) were routinely used. Data are presented as mean ± s.e.m. Mean comparisons were obtained using the unpaired $t$-test or the non parametric Mann–Whitney test. The number of responsive cells in the two conditions was compared using the $\chi^2$ test. The significance level was set for $p < 0.05$.

## RESULTS

### Neuronal differentiation of neuroblastoma F-11 cells

After 12–14 days in 1% FBS medium, F-11 cells stained positively for the neuronal nuclear protein NeuN (Fig. 1) and about 50% of the culture was characterized by neuronal networks of cells exhibiting typical neuronal morphology. When 1% FBS cultures were analyzed by the patch-clamp technique, only cells with neuronal morphology showed electrophysiological properties characteristic of mature neurons (Fig. 2). These cells, defined as "differentiated cells" throughout the article, compared to cells maintained in 10% FBS medium ("undifferentiated cells"), had more hyperpolarized resting membrane potentials ($V_{rest}$: −50.5 ± 1.9 mV vs. −17.1 ± 3.8 mV), and exhibited increased sodium and

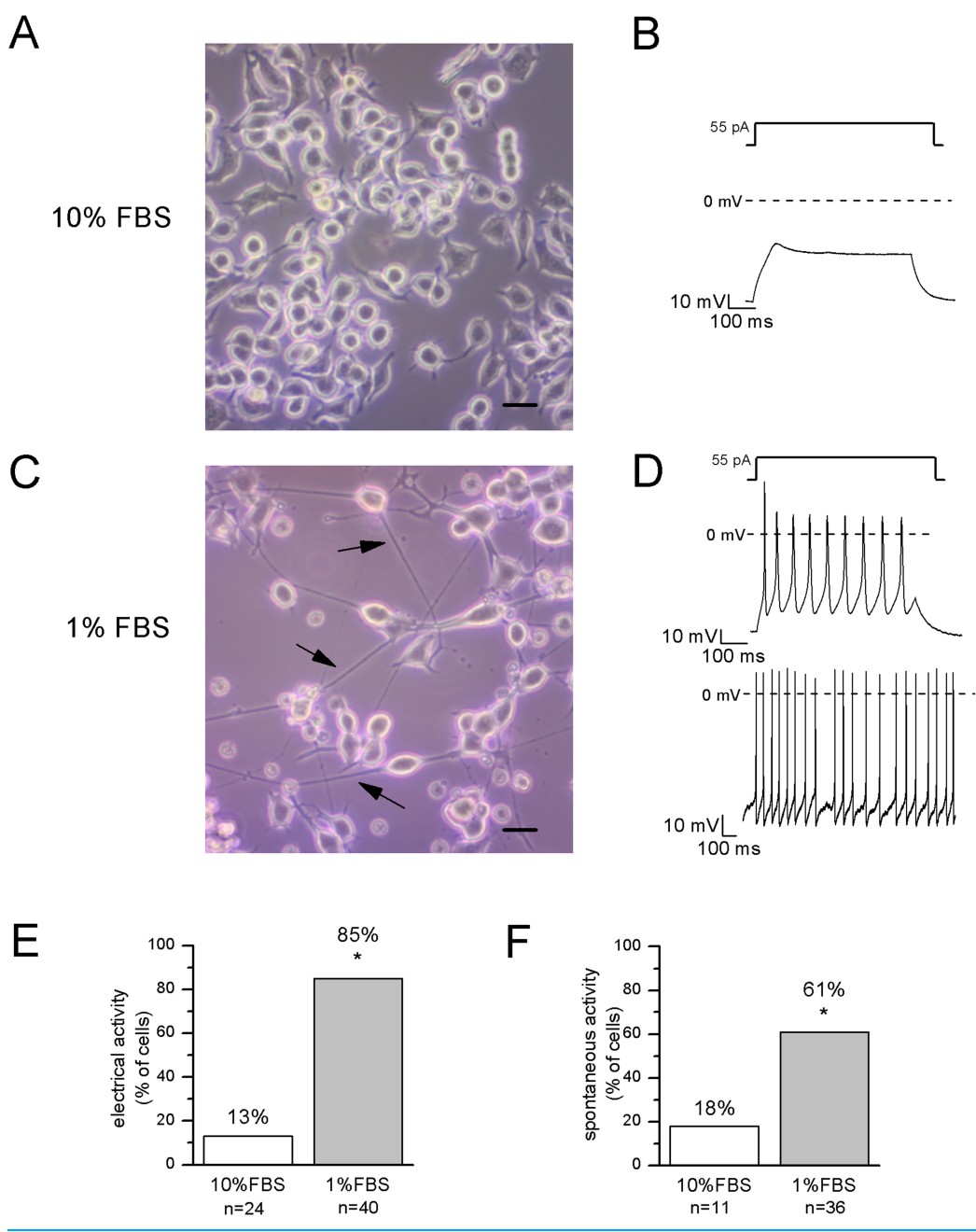

**Figure 2 Differentiated cells with neuronal morphology were selected for electrophysiological recordings.** (A, B) In undifferentiated F-11 cells, the round cell bodies and the absence of neuronal processes were consistent with the lack of electrical activity. Scale bar, 20 µm. (C, D) Differentiated F-11 cells showed oval cell bodies and long processes (indicated by arrows) which were consistent with the discharge of spontaneous or induced action potentials. Scale bar, 20 µm. (E) A significantly higher percentage of differentiated cells was able to fire action potentials compared to undifferentiated cells. (F) Moreover, cells able to generate spontaneous spiking were significantly more represented in the differentiated culture. Asterisks represent significance.

potassium current densities (for $I_{Na}$: 114 ± 10.2 pA/pF vs. 42.5 ± 15 pA/pF; for $I_K$: 181.4 ± 17.9 pA/pF vs. 40.9 ± 5.5 pA/pF). Moreover, a significantly higher percentage of cells was able to fire induced or spontaneous APs. Cells endowed with APs were 85% in

differentiating conditions vs. 13% in control conditions ($\chi^2$ test); moreover cells with spontaneous spiking reached 61% vs. 18% ($\chi^2$ test) (Figs. 2E and 2F). Therefore, we investigated in the differentiated cells the presence of ion channels expressed in DRG neurons.

## Expression of voltage-dependent sodium and potassium channels in differentiated cells

Sodium currents were fast and completely blocked by 1 μM TTX, indicating that differentiated F-11 cells did not express TTX-resistant sodium currents which are conversely present in some classes of DRG neurons. Activation and inactivation properties were consistent with those of TTX-sensitive currents characterized in small DRG neurons by *Cummins & Waxman (1997)* (for activation: $V_{1/2}$ = −22 ± 0.5 mV, $k$ = 6.2 ± 0.4 mV, $n$ = 5; for inactivation: $V_{1/2}$ = −68 ± 2 mV, $k$ = 5 ± 1 mV, $n$ = 7) (Figs. 3A and 3B). Potassium current kinetic and voltage-dependence (Fig. 3A) were consistent with delayed rectifier potassium currents. Potassium current amplitude was reduced of 84% ± 1% by 10 mM TEA administration ($n$ = 17). F-11 cells also expressed ERG potassium current $I_{erg}$ (Figs. 3E–3G), as already referred for undifferentiated F-11 cells in *Faravelli et al. (1996)* and for cells differentiated in retinoic acid by *Chiesa et al. (1997)*. In our conditions, $I_{erg}$ current density significantly increased in differentiated compared to undifferentiated cells (42 ± 9 pA/pF, $n$ = 8, vs. 14 ± 2 pA/pF, $n$ = 14). Thus voltage-dependence of activation and inactivation were also compared. $V_{1/2}$ and $k$ values for activation and inactivation in undifferentiated cells were: −29.7 ± 2.4 and 9 mV ($n$ = 6); −64.8 ± 4.4 and 18 mV ($n$ = 7), respectively. Activation properties did not change in differentiated cells and were $V_{1/2}$ = −32 ± 3 mV and $k$ = 5 mV, $n$ = 12. Concerning the voltage-dependence of inactivation, $V_{1/2}$ was −56 ± 3 mV and $k$ was 12 mV, $n$ = 4. $I_{erg}$ currents from both undifferentiated and differentiated cells were almost completely inhibited by WAY-123,398 (block fractions were respectively, 77% ± 4%, $n$ = 6, and 71% ± 6%, $n$ = 9).

Moreover, they were also blocked by 5 mM $Cs^{2+}$ (mean inhibition was 70% for both undifferentiated and differentiated cells). $Cs^{2+}$ had no effects on the outward potassium components, which conversely were almost completely inhibited by 10 mM TEA (Fig. 3H).

## Barium currents through voltage-dependent calcium channels

In DRG neurons all the different types of voltage-dependent calcium channels have been described, but low voltage-activated calcium currents ($I_{Ca(low)}$) have been identified only in small and medium diameter cells (*Scroggs & Fox, 1992*). To verify the functional expression of calcium channels in F-11 cells, whole-cell $Ba^{2+}$ currents were recorded under conditions which suppressed $Na^+$ and $K^+$ currents, by adding TTX and TEA in the extracellular solution, and CsCl in the patch-pipette (see Materials and Methods for details). The I–V curve was determined by measuring the peak current evoked at potentials from −60 and +50 mV. It showed a peak between −20 and −10 mV in undifferentiated cells ($n$ = 9) and around −10 mV in differentiated cells ($n$ = 8) (Figs. 4A and 4B). To discriminate between high threshold ($I_{Ba(high)}$) and low threshold-activated currents ($I_{Ba(low)}$), two different protocols were applied as described in Materials and Methods.

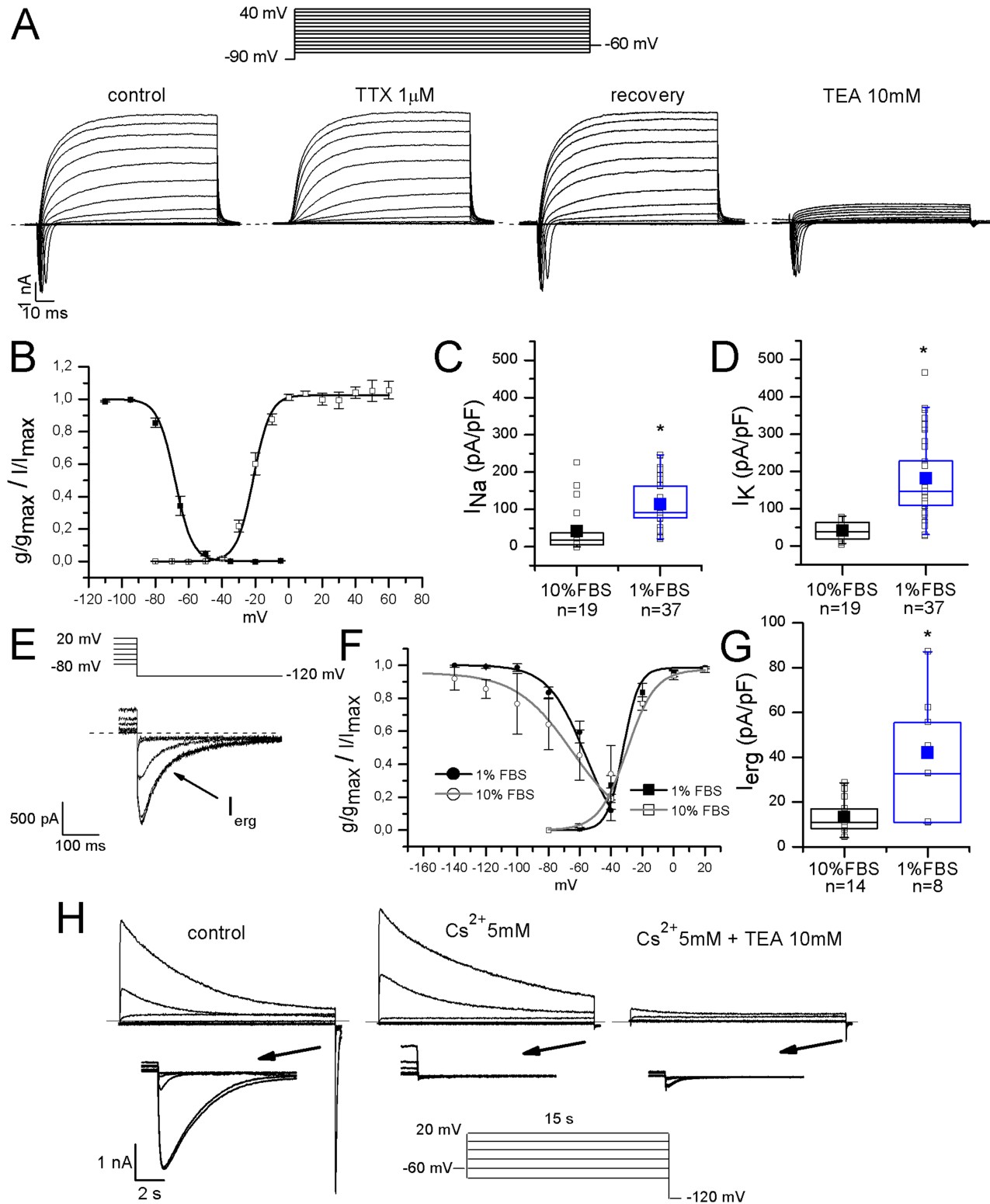

**Figure 3 Differentiated F-11 cells expressed voltage-dependent sodium and potassium currents.** (A) Sodium and potassium currents evoked by depolarizing steps from a preconditioning potential of −90 mV. Sodium currents were isolated by the application of the selective blocker TTX. All sodium currents were TTX-sensitive. Potassium outward currents exhibited properties consistent with delayed rectifier currents and they were inhibited by 10 mM TEA. (B) Activation (g/g$_{max}$, empty square symbols) and inactivation (I/I$_{max}$, filled square symbols) properties of

**PeerJ** ________________________________________________________

**Figure 3 (continued)**

voltage-dependent sodium channels. For activation, $V_{1/2} = -22 \pm 0.5$ mV, $k = 6.2 \pm 0.4$ mV ($n = 5$); for inactivation: $V_{1/2} = -68 \pm 2$ mV, $k = 5 \pm 1$ mV ($n = 7$). (C, D) Sodium ($I_{Na}$) and potassium ($I_K$) current densities in undifferentiated and differentiated F-11 cells. Bar graphs were overlaid with scatter plots. Both $I_{Na}$ and $I_K$ densities were significantly higher in differentiated cells. (E–G) ERG potassium current ($I_{erg}$) density increased in differentiated cells compared to undifferentiated cells ($42 \pm 9$ pA/pF, $n = 8$, vs. $14 \pm 2$ pA/pF, $n = 14$) but the biophysical properties of activation ($I/I_{max}$, square symbols) were not different in differentiated cells compared to undifferentiated cells ($V_{1/2} = -32 \pm 3$ mV and $k = 5$ mV ($n = 12$) for differentiated cells vs. $V_{1/2} = -29.7 \pm 2.4$ mV and $k = 9$ mV ($n = 6$) for undifferentiated cells). Instead, the voltage-dependence of inactivation ($g/g_{max}$, round symbols), was ~8 mV more depolarized in differentiated cells ($V_{1/2} = -56 \pm 3$ mV and $k = 12$ mV ($n = 4$) for differentiated cells vs. $V_{1/2} = -64.8 \pm 4.4$ mV and $k = 18$ mV ($n = 7$) for undifferentiated cells). Extracellular potassium concentration in these experiments was 40 mM. (H) Potassium current sensitivity to $Cs^{2+}$ and TEA block. As represented in the middle panel, inward currents showed sensitivity to $Cs^{2+}$ (mean inhibition was $70\% \pm 5\%$, $n = 15$). When TEA was administered with $Cs^{2+}$ (right panel), outward currents were also blocked. Mean inhibition by TEA was $76\% \pm 2\%$, $n = 15$. The arrows indicate the enlargement of the tail currents evoked at $-120$ mV in the different conditions (control, during $Cs^{2+}$ and during $Cs^{2+}$ and TEA coapplication). Extracellular potassium concentration was 40 mM. Asterisks represent significance.

Test at 0 mV from a holding potential of $-80$ mV could evoke $Ba^{2+}$ currents in 15 out of 21 undifferentiated cells (74%); the mean current amplitude was $107 \pm 26$ pA (current density $3 \pm 0.4$ pA/pF) (Fig. 4C). In differentiated cells currents had mean amplitude of $203 \pm 44$ pA (current density $5 \pm 1$ pA/pF, recorded in 16 out of 24 cells, Fig. 4C). Test at $-50$ mV from a holding potential of $-90$ mV evoked responses in neither undifferentiated nor differentiated cells (13 and 10 cells tested, respectively), indicating that low voltage-activated $Ca^{2+}$ channels were not expressed. Cadmium application (200 μM) completely blocked $I_{Ba}$ in all the differentiated cells ($n = 10$) and in all the undifferentiated cells ($n = 11$). The L-type voltage-gated calcium channel antagonist nifedipine (5 μM) blocked high threshold currents equivalently in differentiated (63% ± 7% of block, 7 out of 7 cells) and undifferentiated cells (51% ± 11% of block, 5 out of 5 cells) (Figs. 4D–4F).

## Capsaicin

Capsaicin, the pungent ingredient of the hot chili pepper, is the agonist of the transient receptor potential vanilloid 1 (TRPV1) non-selective cation channel, highly expressed in DRG sensory neurons (*Goswami et al., 2006*; *Masuoka et al., 2017*). In undifferentiated cells, 3 μM CAPS did not evoke any response ($n = 12$). In differentiated cells, it induced appreciable inward currents (≥20 pA) in 13 out of 62 cells (21%). Mean current amplitude was $41 \pm 9$ pA (current density: $1.2 \pm 0.3$ pA/pF, $n = 13$). The effects of CAPS are shown in Figs. 5A–5E. In our study, no correlation between responses and cell capacitance was evident (mean capacitance of responsive cells: $38.2 \pm 2.7$ pF; mean capacitance of non-responsive cells: $38.2 \pm 4.4$ pF).

## Substance P

Substance P modulates the excitability of sensory neurons in pain pathways. In DRG neurons it can increase or decrease excitability, by modulating ligand-gated channels including P2X3 ATP receptors, TRPV1 CAPS receptors and ASIC3 channels, as well as several types of voltage-gated channels (sodium, calcium and potassium channels, and the hyperpolarization-activated $I_h$ current). In our experiments, undifferentiated cells did not respond to SP ($n = 12$). On the contrary, in differentiated cells, SP depolarized the cells

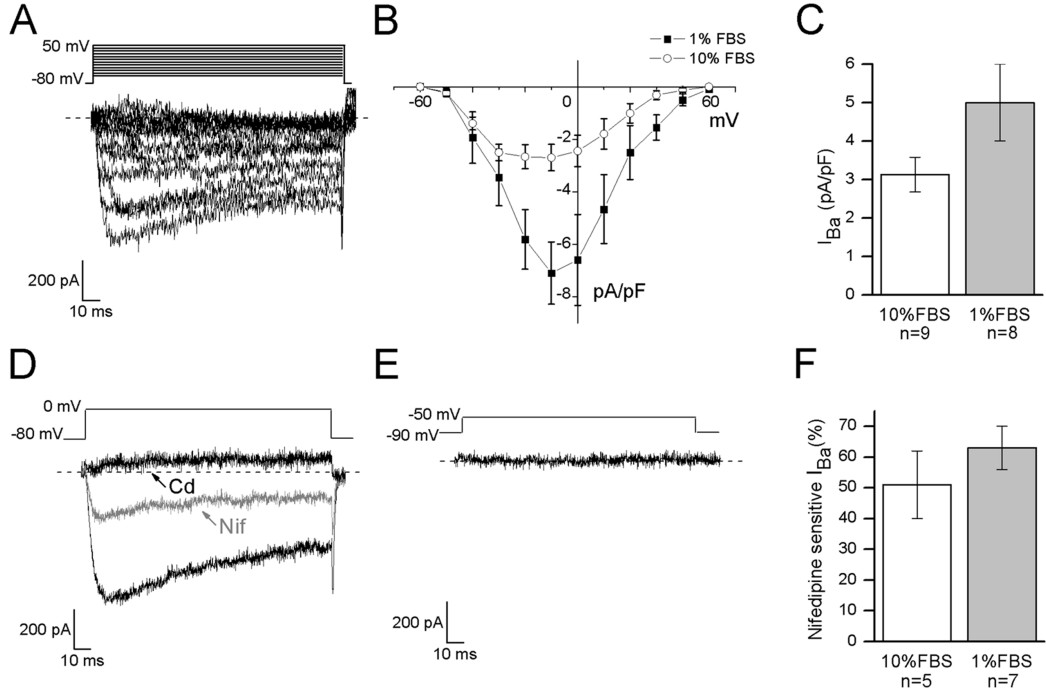

**Figure 4 Differentiated cells expressed high voltage-activated barium currents.** (A, B) Depolarized test potentials (from −60 to 50 mV) from a holding potential of −80 mV evoked barium currents with an I–V relationship which peaked around −10 mV in differentiated cells and between −20 and −10 mV in undifferentiated cells. (C) Current densities showed a tendency to increase in differentiated compared to undifferentiated cells (5 ± 1 pA/pF, $n$ = 16 cells, vs. 3 ± 0.4 pA/pF, $n$ = 15 cells, respectively). (D) To define the contribution of high threshold ($I_{Ba(high)}$) and low threshold ($I_{Ba(low)}$) activated channels the holding and the test potentials were varied opportunely. In both the culture conditions, test potential at 0 mV from a holding of −80 mV evoked currents, which were completely blocked by cadmium application (Cd, 200 μM) and partially inhibited by nifedipine (Nif, 5 μM). (E) On the contrary, no currents were elicited by testing at −50 mV from a holding potential of −90 mV, demonstrating that low threshold-activated $Ca^{2+}$ channels were not present in cell membranes ($n$ = 13 undifferentiated cells and $n$ = 10 differentiated cells). (F) As shown in the histograms, nifedipine-sensitive currents were equivalently expressed in both the culture conditions.

of about 12 mV (from −48 ± 4 to −36 ± 4 mV, $n$ = 9). No cell tested underwent hyperpolarization in presence of the substance. When tested in the voltage-clamp mode, SP promoted small inward currents in 11 cells. Mean current amplitude calculated for currents ≥20 pA was 24.5 ± 3.3 pA ($n$ = 4) (Figs. 5F–5J). Responses to SP were recorded in 79% of cells (15 out of 19) overall. No correlation between responses to SP and cell capacitance was evident.

## Acidic solutions

Acid-sensing channels are expressed by neurons throughout the nervous system and are involved in acidotoxicity related to several pathological conditions and the perception of pain. F-11 cells maintained at holding potential of −70 mV responded to the application of acidic (pH 5 and pH 6) solutions with fast inward currents (Figs. 5K–5T) which were recorded in <40% of undifferentiated cells (pH 6: 31%, 5 out of 16 cells,

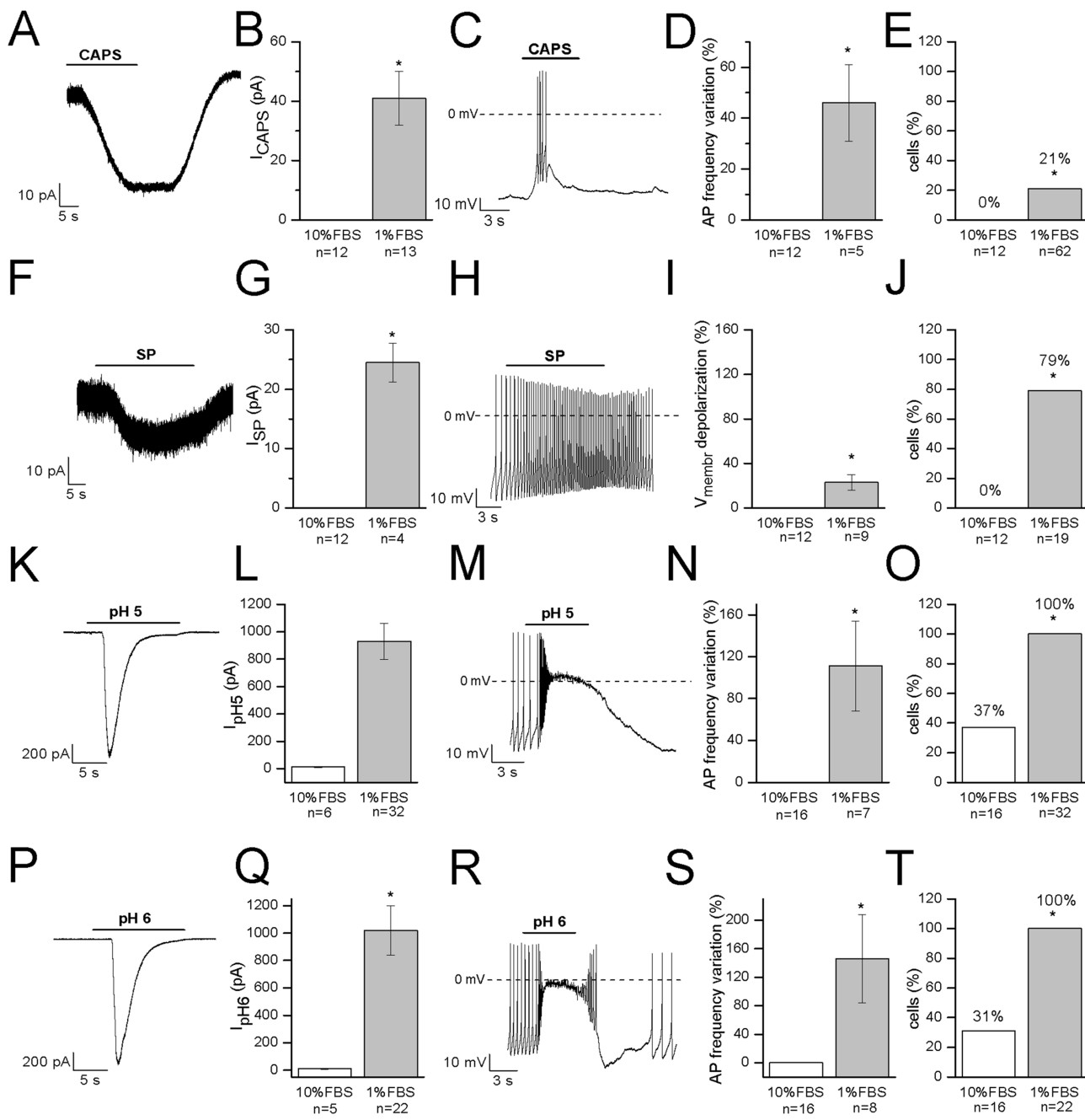

**Figure 5 Differentiated F-11 cells express receptors and ion channels of nociceptors.** The representative currents and the effects on the electrical activity evoked by capsaicin, substance P and acidic pH values are shown. (A–E) Capsaicin (CAPS) evoked responses in differentiated but not in undifferentiated cells. (F–J) Substance P (SP) induced currents and high frequency action potential discharges in differentiated cells but had no effect on undifferentiated cells. Responses to acidic extracellular solutions, (K–O) pH 5 and (P–T) pH 6, were recorded in all the differentiated cells and in 31% and 37% of undifferentiated cells, respectively. Cell membrane potential was clamped at −70 mV during all the experiments. Asterisks represent significance.

mean current: 8.6 ± 1.8 pA; pH 5: 37%, 6 out of 16 cells, mean current: 11.6 ± 2.8 pA). On the contrary, they were evoked in all the differentiated cells with a mean current amplitude of 1,021 ± 181 pA ($n = 22$) at pH 6 and of 931 ± 131 pA ($n = 32$) at pH 5.

## Acetylcholine and nicotinic acetylcholine receptors

Functional nicotinic acetylcholine receptors (nAChRs) have been described in heterogeneous populations of dissociated rat and mouse DRG neurons (*Smith et al., 2013*) and they are known to be involved in pain modulation. The pathway in which nAChRs operate to modulate pain is actually of great interest since it has been suggested that the anti-allodynic effect of their agonists may have a peripheral component (*Rueter et al., 2003*).

In undifferentiated cells, ACh evoked currents in 14 out of 14 cells, with mean amplitude 43.6 ± 12 pA at −70 mV. In differentiated F-11 cells, currents were recorded in 89% of cells and displayed a significantly higher mean amplitude (136 ± 35 pA, $n = 34$ out of 38). In differentiated cells the effects of ACh administration was also evident on the resting membrane potential (mean depolarization: +44% ± 8%, $n = 20$). Since ACh-evoked currents were completely inhibited by 1 μM DTC, they were consistent with nAChRs (Figs. 6A–6D). Approximately the same percentage (70–80%) of rat DRG primary neurons was referred to express functional nAChRs in *Genzen, Van Cleve & McGehee (2001)*.

## Glutamate receptors

The localization of AMPA, Kainate, and NMDA receptor subunits has been demonstrated in rat DRGs by immunohistochemistry and in situ hybridization histochemistry, suggesting that the glutamatergic system plays an important role in the primary sensory afferent systems (*Sato et al., 1993*). In our experiments 1 mM glutamate was effective on only 1 out of 13 undifferentiated cells, but on 28 out of 33 differentiated cells (85%). In the current-clamp mode, it depolarized differentiated cells of 27% ± 5% ($n = 14$), and in voltage-clamp recordings it evoked currents with amplitude ranging from 20 to 227 pA (mean amplitude: 77 ± 25 pA, $n = 10$) (Figs. 6E–6H). Currents were inhibited by 71% ± 9% during CNQX and AP5 co-application. However, glutamate administration in the $Mg^{2+}$-free extracellular solution and at the holding potential of −40 mV evoked no AP5-sensitive currents, demonstrating that differentiated F-11 cells expressed non-NMDA receptors prevalently.

## DISCUSSION

The DRG neuron-derived immortal cell line F-11 is routinely used as in vitro model of peripheral sensory neurons. However, expression analysis of RNA transcripts using next-generation sequencing (*Yin, Baillie & Vetter, 2016*) has stressed the need for the exploration on the functional receptors they present to validate this cell line as a model of DRG neurons. In this paper, for the first time, we show an extensive characterization by the patch-clamp technique of the functional properties of this cell line, comparing undifferentiated cells (7 days in 10% FBS medium) with the ones differentiated by a very simple and economic procedure, represented by cell incubation in serum deprived medium for 10–14 days in culture.

Undifferentiated cells showed immature neuronal morphology and behavior, with low expression of voltage-dependent channels and reduced capability of generating APs, and did not react to CAPS and SP administration. In contrast, differentiated cells revealed typical features of neurons (long processes and NeuN expression) and, when analyzed by

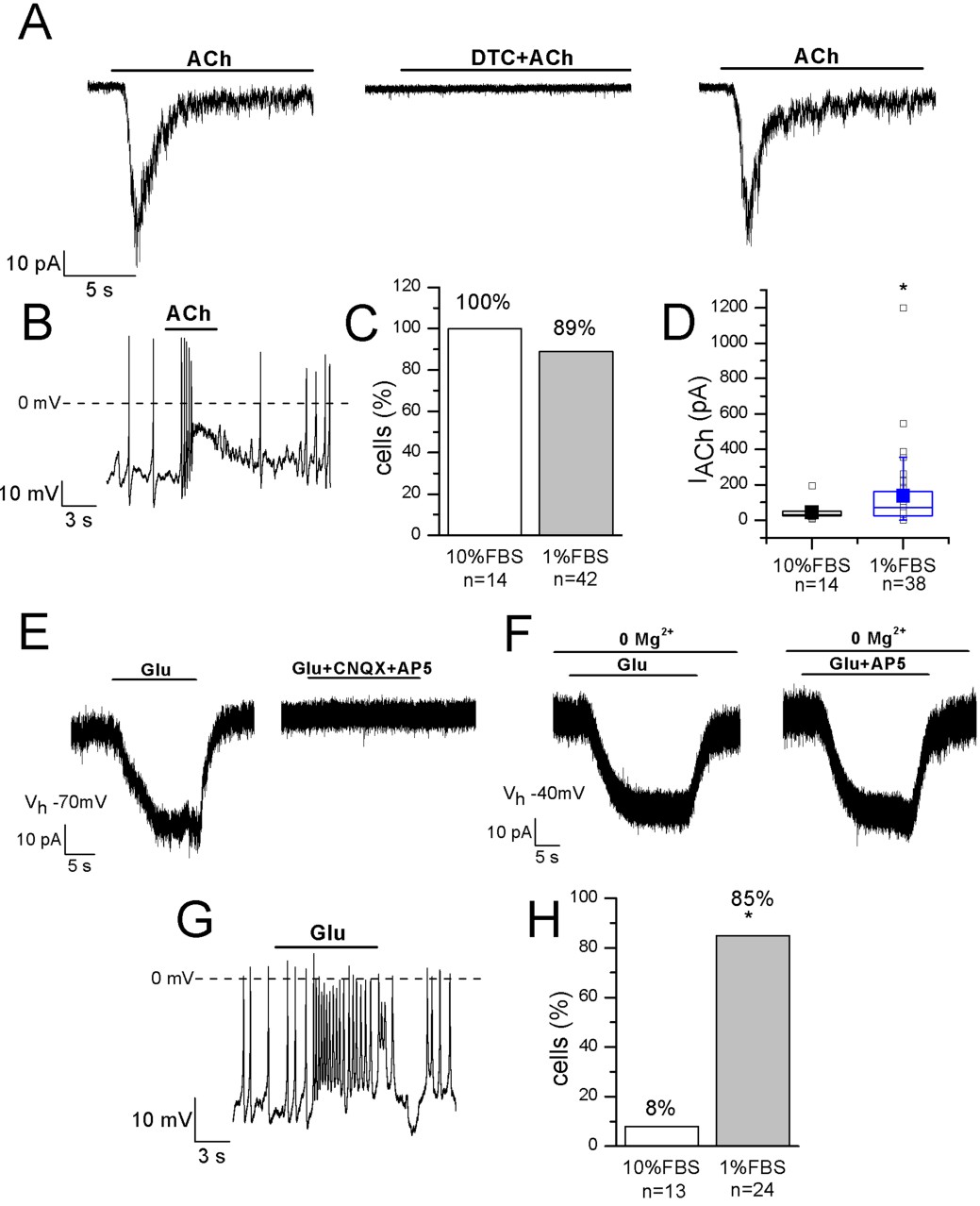

**Figure 6 Differentiated F-11 cells display responses to acetylcholine and glutamate.** The representative currents and the effects on the electrical activity evoked by acetylcholine (ACh) and glutamate (Glu) are shown. (A–D) ACh was effective on 100% of undifferentiated and on 89% of differentiated cells and its action was mediated by nAChRs, as demonstrated by d-tubocurarine (DTC) block. Cell membrane potential was clamped at −70 mV. (E–H) On the contrary, the percentage of Glu-responsive cells was significantly higher in differentiated than in undifferentiated cells. Glu-evoked effects were principally mediated by non-NMDA receptors, since AP5 administration in $Mg^{2+}$-free extracellular solution and at −40 mV did not affected them. Asterisks represent significance.

the patch-clamp technique, they were functional, firing APs spontaneously or after current injection, and expressing voltage-gated sodium, potassium, and calcium channels. Sodium currents evoked in differentiated F-11 cells were consistent with those exhibited by

primary DRG neurons, although TTX-resistant currents typical of nociceptors were not detected. Literature refers that in dissociated rat DRG neurons, calcium currents and calcium transients are sustained by different voltage-dependent calcium channels (N-, P/Q-, R-type, and T-type channels), whose variable expression was related to different cell body diameters; currents through N-type channels instead remained constant between the diameter ranges (*Scroggs & Fox, 1992*; *Fuchs et al., 2007*). In differentiated F-11 cells, the largest barium current activated at 0 mV, and was consistent with high threshold activated $Ca^{2+}$ channel subtypes, whereas no current was attributed to low threshold activated channels. The absence of low threshold activated (T-type channels) $Ca^{2+}$ currents and the expression of TTX-sensitive $Na^+$ channels in the differentiated F-11 cells of our study correlated with the generation of a regular tonic firing of APs, as it has been described for afferent TTX-sensitive A fibers innervating the proximal urethra and the bladder (*Yoshimura et al., 2003*).

It is known that voltage-gated currents recorded from neurons are distorted due to the lack of space clamp and in fact the results published on DRG neurons are often obtained from cells without processes. In the simulations of all the neuronal morphologies, even of neurons with relatively short dendrites, the membrane potential imposed at the soma decayed by ~10–20 mV over the first 100 µm along the dendrite away from the somatic voltage-clamp (*Bar-Yehuda & Korngreen, 2008*). However, since indications from both morphology and function are needed to define the level of neuronal maturation, in this manuscript we confirmed that cells displaying elongated processes expressed the typical electrical activity of mature neurons. Even if we are aware of the limits of our analysis and of the distortion of the biophysical properties we described for $Na^+$, $K^+$, and $Ca^{2+}$ channels, nevertheless we show that this properties sustain a neuronal behavior which is consistent with the one described for primary sensory neurons (see Table S1 for the electrophysiological properties we described in undifferentiated and differentiated F-11 cells, and for the same properties referred in literature for primary DRG neurons and for other sensory neuron models for comparison).

Concerning ligand-gated channels, we verified differentiated F-11 cell sensitivity to ACh since nAChRs are expressed on rat DRG neurons (*Genzen, Van Cleve & McGehee, 2001*) with a role in pain modulation (*Rueter et al., 2003*). In differentiated F-11 cells, ACh evoked responses in 89% of recorded cells, consistent with the percentage referred by *Genzen, Van Cleve & McGehee (2001)* in rat DRG primary neurons. The complete inhibition by DTC confirmed that ACh-evoked currents were sustained by nAChRs. We also investigated the effects of glutamate, since it has been suggested that glutamatergic system plays an important role in the primary sensory afferent pathway (*Sato et al., 1993*). In fact, the localization of AMPA, Kainate and NMDA receptor subunits has been demonstrated in rat DRGs, in the peripheral axons of small diameter fibers in the rat and human skin, and in the peripheral terminals of primary afferent nerves innervating somatic tissues (*Sato et al., 1993*; *Coggeshall & Carlton, 1997*; *Carlton & Coggeshall, 1999*; *Kinkelin et al., 2000*). Moreover, activation of non-NMDA receptors has been shown to modify the electrical activity of embryonic DRG neurons (*Lee et al., 2004*). In our experiments 1 mM glutamate was predominantly effective on non-NMDA receptors.

Capsaicin sensitivity is a hallmark of nociceptive sensory neurons and we investigated its effect, together with the action of SP, on differentiated F-11 cells. CAPS can have both irritating and analgesic effects (*Fitzgerald, 1983*). It is the agonist of the transient receptor potential vanilloid 1 (TRPV1) non-selective cation channel, a polymodal sensor sensitive to heat, acid pH, and irritant vanilloids, highly expressed in a subset of DRG sensory neurons (*Goswami et al., 2006*; *Masuoka et al., 2017*). The painful sensation induced by CAPS is consequent to its binding to TRPV1, and to $Ca^{2+}$ and cation influx through them, which activates several mechanisms (*Takayama et al., 2015*; *Frias & Merighi, 2016*; *Goswami et al., 2006*). Although very small amounts of TRPV1 transcript were identified in proliferating F-11 cells by *Yin, Baillie & Vetter (2016)*, patch-clamp recordings revealed that CAPS was able to evoke calcium currents in roughly 30% of the cells examined (*Kusano & Gainer, 1993*). In our conditions, differentiation induced responses to CAPS in 21% of cells, a percentage consistent with the one referred by Kusano & Gainer, but inferior to the one found by *Ambrosino et al. (2013)* in F-11 cells differentiated in retinoic acid.

Substance P is released by DRG neurons at regions in the CNS associated with pain transmission, and at the periphery, where it contributes to neurogenic inflammation in many tissues. Moreover, it can increase or decrease excitability of sensory neurons, by modulating various ligand- and voltage-gated channels, depending on cell diameter and on the time course of AP after-hyperpolarization (*Moraes, Kushmerick & Naves, 2014*). In our experiments, responses to SP were recorded in 79% of cells. When investigations were performed in the voltage-clamp mode, SP evoked small inward currents. In the current-clamp mode, administration of SP depolarized the cells. Contrary to rat primary DRG neurons (*Moraes, Kushmerick & Naves, 2014*), responses to CAPS and SP coexisted in the same differentiated F-11 cell.

Dorsal root ganglion neurons are also sensitive to variation in extracellular pH because of the proton-activated cation channels they express. In particular, two principal proton-gated inward currents were recorded from them and described in literature: fast and rapidly inactivating currents, with maximum activation at pH 6, and sustained, slowly inactivating currents, activated only at pH below 6.2, observed exclusively in DRG neurons responsive to CAPS (*Bevan & Yeats, 1991*). In our work, proton activated currents were recorded in 100% of the differentiated cells we tested.

Although F-11 cells have been classically used as a model of authentic type C peptidergic nociceptive neurons, for their ability to synthesize and secrete SP, to express sensory neuron antigens, functional voltage-dependent calcium channels and CAPS receptor TRPV1 (*Francel et al., 1987*; *Boland & Dingledine, 1990*; *Jahnel et al., 2001*), our experiments confirm the heterogeneity of these cells already hypothesized by *Kusano & Gainer (1993)*. Since in the past years different culture conditions have been used to attain F-11 cell differentiation, it is possible that these conditions are responsible for the heterogeneous characteristics described in literature.

Referring to the sub-classification of acutely dissociated cells of rat DRGs, which was proposed by *Petruska et al. (2000)* by using a "current signature" method based on the algesic profile (responses to pH5 and sensitivity to both ATP and CAPS), in our study it

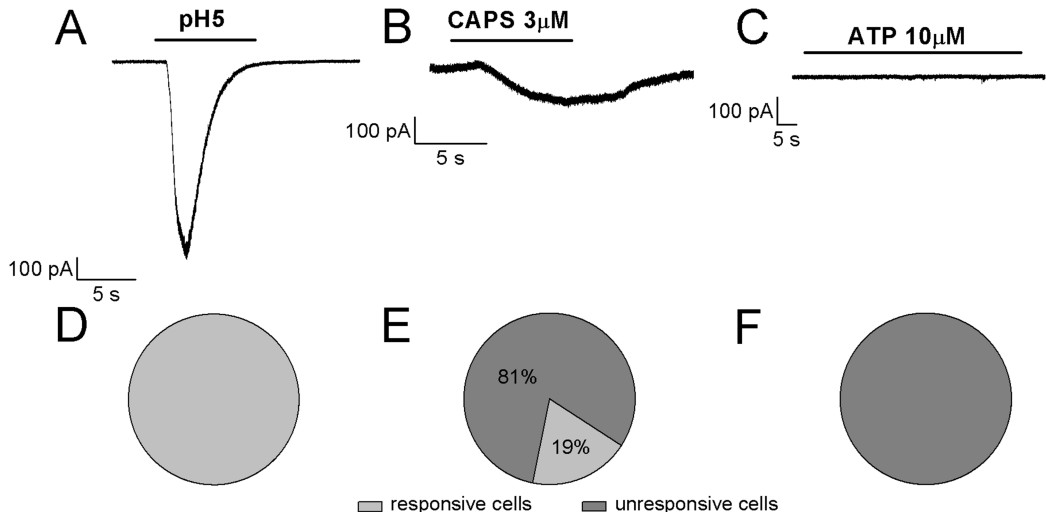

**Figure 7 Algesic profile of differentiated F-11 cells.** Responses to (A) pH5, (B) ATP, and (C) capsaicin (CAPS) were investigated in differentiated cells to define the "current signature" used by *Petruska et al. (2000)* to subclassify acutely dissociated cells of rat DRGs. (D–F) While all the cells responded to an extracellular acidic solution at pH 5, no cells showed appreciable sensitivity to ATP and only a small fraction of cells (19%) responded to CAPS.

seems that ~80% of differentiated F-11 cells showed similarities with "type 3," CAPS and ATP insensitive cells, whereas ~20% of cells seemed to show a partial correspondence with "type 7," CAPS and ATP weakly sensitive cells. As shown in Fig. 7, the cells stimulated by pH5 responded with desensitizing currents, which were in the 50% of the cells inhibited by 100 μM amiloride (maximum block was 89%; mean block for six cells was 57%, $n = 6$). None of the cells responded at 10 μM ATP administration with appreciable currents, whereas 19% of cells responded to 3 μM CAPS.

## CONCLUSIONS

Dissociated human DRGs represent the ideal model for investigating sensory neurons and the molecular mechanisms of pain (*Valeyev et al., 1996*; *Dib-Hajj et al., 1999*: *Davidson et al., 2014*; *Zhang et al., 2017*). However, their limited availability and the incomplete characterization of ion channel expression and biophysical properties force researchers to make do with rodent DRG neurons, even if the obtained results are controversial. Cell lines are also a debated alternative. In this paper, we show that serum deprived differentiated F-11 cells express some ion channels described in sensory neurons. Moreover, compared to neurons differentiated from immortalized human DRG by *Raymon et al. (1999)* they represent a more accessible model, simple and less expensive. In conclusion, differentiated F-11 cells represent a useful model for research on DRG neurons and, since they express some ion channels and receptors that are also expressed in sensory neurons, might be employed for studying mechanisms involved in the detection and transmission of noxious stimuli.

## ACKNOWLEDGEMENTS

We thank Dr. Michela Ceriani for immunofluorescence analysis supervision.

### Funding

This work was funded by Fondo di Ateneo per la Ricerca (FARgrant) from the University of Milano-Bicocca and by Fondo per il finanziamento delle attività base di ricerca (FFABR) from Italian Ministry of Education, University and Research to Marzia Lecchi. The funders had no role in study design, data collection and analysis, decision to publish, or preparation of the manuscript.

### Grant Disclosures

The following grant information was disclosed by the authors:
Fondo di Ateneo per la Ricerca (FARgrant) from the University of Milano-Bicocca and by Fondo per il finanziamento delle attività base di ricerca (FFABR) from Italian Ministry of Education, University and Research to Marzia Lecchi.

### Competing Interests

The authors declare that they have no competing interests.

### Author Contributions

- Valentina Pastori conceived and designed the experiments, performed the experiments, analyzed the data, prepared figures and/or tables, authored or reviewed drafts of the paper, approved the final draft, performed electrophysiological recordings.
- Alessia D'Aloia performed the experiments, prepared figures and/or tables, approved the final draft, designed and performed immunofluorescence analysis.
- Stefania Blasa performed the experiments, authored or reviewed drafts of the paper, approved the final draft, performed electrophysiological recordings.
- Marzia Lecchi conceived and designed the experiments, analyzed the data, contributed reagents/materials/analysis tools, prepared figures and/or tables, authored and reviewed drafts of the paper, approved the final draft.

### Data Availability

Data are available in the Supplemental Files.

### Supplemental Information

Supplemental information for this article can be found online at http://dx.doi.org/10.7717/peerj.7951#supplemental-information.

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
