# Peer review of "Serum-deprived differentiated neuroblastoma F-11 cells express functional dorsal root ganglion neuron properties"

_PeerJ, doi:10.7717/peerj.7951_

## Round 0.1 · original submission · Major Revisions

This paper explores the properties of a rat/mouse embryonic dorsal root ganglion/neuroblastoma hybrid cell line grown in low serum. While there are good ethical reasons to reduce the number of animals killed for research, and some concerns about rodent/human translatability, it is not clear to me that F11 cells will ever be an answer. It is true that the half-sibling cell line NG108-15 (which shares the same neuroblastoma parent) taught us a lot 20-30 years ago about ion channels and receptors, but times have moved on.

This paper does a reasonable job of describing some of the properties of differentiated F11 cells that might make them a useable model of a sensory neuron, but as pointed out by the Reviewers, there are a number of areas that need attention. Any revised Ms needs to address all the comments made, and particular attention needs to be paid to reporting the number of cells recorded, and the number of different passages used.

The authors should also justify their use of statistics (I mean, use of them at all). There is no indication that the cells in the 2 conditions are sampled either randomly, or purposefully using the same criteria. If neither of these are done, then statistics are probably not appropriate. If the authors can justify their use, then they should pick a level at which they consider a statistical difference is appropriate (P < 0.05 or 0.01 or whatever) and then simply that the values of a given parameter did or did not meet this threshold, rather than reporting p < 10-8 or similar. I think that the differences between the 2 groups are clear enough without the use of statistics.

In addition, the authors should also address the following:

The immunohistochemistry is not quantified, and there are no antibody controls presented. No evidence is provided that the levels are beta-tubulin are different between the 2 cells groups. This should be done or the Figure dropped. It might be more useful to stain for markers associated with sensory neurons such as NF-200 or NeuN.

The picture in Figure 2 shows the different morphology of the cells grown in 10% and 1% serum, that is probably all that is necessary. The authors should more explicitly describe how they chose cells for recording, particularly in the “differentiated state” – if cells were not chosen at random in both low and normal serum, it is probably not appropriate to do statistics at all. There are plenty of cells with fewer processes in the serum-starved cells of Figure 1 – did these cells have properties similar to serum starved or cells grown in 10% FBS ?

As indicated, the recordings of calcium channels are not adequately described, and currents not appropriately characterized. At minimum there should be I/V relationships, but toxins or ions should be used to at least broadly separate out channels types. In addition to the reviewers comments, the authors should report what the pipette offset was in the TEACl/CsF solutions, and either indicate whether they were corrected for in the reporting of the Vm. The trace of the LVA current is clearly not clamped appropriately (the kinetics of channel activation has a clear breakpoint, and there is no channel inactivation) – traces including the repolarization of the membrane should be shown (i.e., with tails).

Please supply the catalog number of the serum and DMEM used here. Was the same batch of serum used for all experiments ? Do the authors expect the same results from different batches of serum ?

The authors should describe the Vm at which the experiments with capsaicin, acid, glutamate etc were done. The authors should also consider that if the cells were being held at -40 or below, the Mg in their extracellular solution would substantially any NMDA receptors in the cells, rendering their estimates of NMDA/non-NMDA contributions to the small currents somewhat irrelevant. It would be valuable to more thoroughly establish the channels/receptors responsible for the observed effects of glutamate, SP and acid.

The authors should more clearly describe how they measured the amplitude of the Na and K channels for reporting the current density – it is unclear whether density was measured at a single potential, or whatever the potential that the greatest inward/outward current was ? At what point in the test step was the K channel amplitude measured ?

The authors should compare K channel activation/inactivation values between their differentiated and undifferentiated cells, and not with someone else’s F11 cells (line 158/159).

Line 213 – the authors mention EPSC frequency – there is no indication in the Ms how they were measured or quantified, and no examples are presented. This needs to be removed or explained.

·

Basic reporting

The paper describes a basic electrophysiological characterization of low serum differentiated F11 cells that induces them to fire action potentials and express some channels that are missing in the undifferentiated cells.

Figure 1 is not helpful it should be removed. The images in 2AB are sufficient.

I suggest adding a table that compares the characteristics (voltage-gated and ligand-gated channel responses, Av cell capacitance, etc), including the % responding cells measured in
- high and low serum F11 cells and,
- isolated DRG neurons,
- F11 cells differentiated in other ways and,
- one or two other key sensory-model cell lines.

This will have the advantage of clearly placing this study in the context of published data and remove the need for the many bar graphs reporting the % of responding cells.

Thank you for providing the raw data, Make sure there is a short explanation in the raw data file that describes what each data table is and notes the corresponding figure in the paper.

The article is generally well presented and easy to read, however there are a few sections that require improvements to the phrasing. Be careful particularly when reporting the results as it is sometime becomes difficult to follow.

Line 32, “…the most (significant?) differences (when compared?) with DRGs….
Line 35, Revise, The use of “consistence” is incorrect
Line 143, unclear sentence
Line 146, Unclear
Line 161, “even if low.. “ unclear
Line 190, delete “any”
Line 200, 100% of cells and is all the cells responsive to capsaicin is obsolete
Line 203, What is the “sustained component at pH5”
Line 252, what is the significance of N-type Cav currents?
Line 284-285 unclear section.
Line 293, unclear “It promoted small inward currents and depolarised 23% of differentiated F11 cells"?

Experimental design

- Figure 2 – Include a figure illustrating the resting membrane potential in high and low serum cells. Was only tonic firing patterns observed? Are they compatible with isolated DRGs?
- Figure 3, DRG neurons should not be mentioned in the figure title. Indicate what each figure symbols is. Indicated what the inlay-ed trace under the "TTX-sensitive Na current" is.
- Please show data demonstrating that NavCh are TTX-sensitivity, KvCh (including ERG) are sensitivity to TEACl and CsCl.
- Figure 3E, Confirm these barium currents are inhibited by cadmium. Why hasn’t Fluoride inhibited the IBa(high) current? Are these the best recordings you have? The noise of the left trace and shape of the right trace is unusual. Indicated the resting current.
- Figure 4. Include a bar graph of peak current and a measure of AP change in frequency in 10 and 1% FBS.
- The currents recorded in response to capsaicin and SP are very small (25 and 14pA) – comment on their expression levels and what sort of studies they will be useful for.
- Figure 5, Line 228, you cannot make any conclusions about the relative expression of NMDA and AMPAR from the described experiment. Please show the AP5 sensitive component in figure 5B.

Validity of the findings

The data seems largely consistent with the conclusions. However, it needs to be strengthened by demonstrating that the voltage gated Na+, K+ and Ca2+ channel currents are sensitive to TTX, Cs/TEA and Cadmium respectively. TRPV1, SP and acid- currents should also be confirmed with appropriate antagonist.

Reviewer 2 ·

Basic reporting

This study shows that serum deprivation produces differentiation of F-11 cells which change their membrane electrical properties. The changes increase their similarities to DRG neurons. Although it has been demonstrated in the literature that serum deprivation changes cell properties and may provide strategies to enhance cell lines use, this seems to be the first report of its use in F-11 cells.

The text needs an extensive revision. The abstract needs a special improvement. Some expressions like “to investigate endogenous properties” and “the sub-type of interest” could be replaced by more precise descriptions.

In general, the results section makes many claims without providing the necessary references.

The discussion repeats the results presentation. It would be interesting to discuss, for example, the similarity of the differentiated F-11 cells with specific sub-types of DRG neurons. Also, why and how their findings are relevant for the research in the area.

Experimental design

The experiments were conducted in a small number of cells. It is necessary to state the number of cells in every group, and the number of cultures they used. When reporting the percentage, it would be interesting to report, in the figures, the number of cells that responded to a specific stimulus over the total number of cells tested.

The protocols for current isolation are adequate.

Line 106 WAY-123,398

The complex geometry of the differentiated cells will make voltage clamp more difficult. Literature results from DRG neurons are often based on data from cell bodies with few or no processes, exactly to avoid space clamp issues. Authors should include a paragraph in their discussion evaluating if and how the morphological changes may affect their data.

Line 77 – please provide details for staining with DiI.

Line 128 Excel (Microsoft)

Line 191 – It would be difficult to measure a mean of 14 pA with a noise of about 10 pA (figure 4). Please review this data.

Validity of the findings

The scientific question is clear, although the rationale is not well described.

The rationale for conducting this study is not because DRG neurons are heterogeneous and change their properties in culture, so we need an alternative that also change their properties in culture.

Regarding to heterogeneity, lines 171-173 report the expression of LVA Ca2+ channels in 4 cells tested, and the lack of expression in the other 7 cells. This result indicates heterogeneity in this regard in their system. Although similar heterogeneity is observed in different classes of DRG neurons, what difference may exist in their culture from cell-to-cell? Similar question for the sensitivity to capsaicin (lines 174-183).

Additional comments

line 125 – The expression “to study pain response” referring to the use of capsaicin is inadequate. Pain is a subjective sensation. In animals, the term nociception is used. In isolated cells, potential nociceptive stimulus produces response through nociceptive transducing receptors.
Line 143 “cells with electrical activity” – a more specific parameter would be appropriate.
Line 146 – it is not clear what authors mean by “intrinsic cues”
Line 147 The subtitle needs a complement “Expression of voltage dependent sodium and potassium channels in differentiated cells”
Line 149 present in some classes of DRG neurons
Lines 161-163 are unclear. Does it mean that all calcium currents sub-types are present, but low threshold is only observed in medium and small cells?
Line 168 should state first the % of cells with currents, to compare with control, then give details of mean current density. Use the same criteria to report the variables in different groups.
Line 189 “depolarized the cells by 23±7%”, “No any cell tested…” change to: No cells tested…
Line 193 – “such as for capsaicin” is incorrect and unnecessary
Line 284 – Change “cells which manifest…” to: Differentiation of F11 cells produced 44% increase in response to capsaicin.

---

## Round 0.2 · Minor Revisions

The authors have made substantial revisions and provided new experiments that have improved the manuscript. However, there are still some things that need attention, including comments from Reviewer 1, and those below.

- “increasing trend” should be removed from the abstract – it is not a meaningful scientific term

- the authors compare the properties of their cells with a several types of specific groups of sensory afferents – either bladder afferents or type 3/7 cells from the byzantine subtyping of rat DRG cells that seems to a favoured past-time of some. Presumably the resemblance is superficial (for example, there is no ATP response in F11 cells), and I don’t think these comparisons are helpful – F11 cells are a heterogeneous interspecies hybrid that may be useful for studying the molecular properties of molecules involved in the detection and transmission of some noxious stimuli, but they are not and never will be sensory neurons. I suppose the authors can pretend as they wish, but I think comparisons to actual subtypes of sensory neuron on the basis of what is presented here is specious. But whether the authors reconsider is up to them – I won’t knock back the paper on this basis.

- line 51 – F11 cells do not “perform the functions of nociceptive and non-nociceptive neurons”, they express some proteins that are also expressed in sensory neurons (but not exclusively in sensory neurons – e.g. TRPV1 in the brain).

- line 142 – “addition” not “addiction”

- line 155 – the authors are not studying nociception, but the expression of proteins involved in nociception

- line 162 – the authors set P < 0.05 as significance, yet report P values of 0.01, 0.001 etc throughout the Ms. Please be consistent, set you significance value and report whether parameters were or were not different

- line 193 – the authors should comment on the substantial differences in the biophysical properties of the putative ERG current between the differentiated and undifferentiated cells – the half activation differs by 30 mV – what could the explanation be ?

- line 237 – What does “depolarized the cells by 23 ± 7% mean” ? Probably should just repot the change in Vm.

- The discussion of TRPV1 in sensory neurons is long and probably unnecessary, especially as less than 25% of differentiated F11 cells responded to it.

- Can the authors comment on whether the proteins whose activity they measured in this study are likely to be rat, mouse, or both ? This could be important for channels which form multimeric complexes – like TRPV1 or nACHR.

·

Basic reporting

There are some sections that need improvement.
Please fix typing errors, revise the phrasing/clarity at the following places: Lines 43,48,169,177,212,249 (remove capsaicin line it has no meaning), 293-295, 302,205-14,335,397.
Avoid using phrases like line 51, “perform functions of nociceptive and non-nociceptive sensory neurons”; Line 155, “to study nociception”; Line 386 “…include neurons with nonnociceptive and nociceptive functions respectively”. They are inaccurate.
Please use consistent language around 1 and 10% serum grown F-11 cells. (Differentiated, 1% FBS, high/low serum, serum deprived etc). eg. I don’t think differentiated cells has the same meaning at line 170 vs line 60 & 63.
Line 395 is overstated.

Experimental design

The identity of the cells being sampled and compared in Fig. 2 and throughout the paper still needs to be clarified in the text. The statistics comments about comparison from the editor is valid and should be taken into account.
I believe you only recorded from 1% FBS cells with neuronal morphology and from cells chosen at random in the 10% FBS condition. Are the 1% FBS cells without neuronal morphology different from the ones with neuronal morphology? What proportion of 1% FBS cells have neuronal morphology? This is of interest to anyone who is considering using this model. Report a type of current occurs in x% of sampled cells.
Given NMDA receptors are inhibited by pore Mg2+ most of the time. Without membrane depolarisation/Mg2+ free solutions would you expect to see an AP5 effect even if NMDAR were highly expressed?

Validity of the findings

Stated aim: "characterization of the electrophysiological properties of F-11 cells differentiated by serum deprivation and investigate if differentiated F-11 cells manifest similarities with DRG neurons" This aim was successfully carried out.
The justification was to: "to verify whether differentiated F-11 cells are an adequate model of sensory neurons to be employed in the study of peripheral neuropathy and in the design of strategies of peripheral nerve reconstruction. I think the differentiated F-11 cells have more DRG-like properties than undifferentiated F-11 cells. I can not see how they would be helpful to study periperal neuropathy and nerve reconstruction design. If true, this should be explained in the discussion.

The paper describes a simple method to make the characteristics of F-11 cells more “DRG-like”. Like previously described differentiation techniques, they could be of use if a researcher is investigating some types of voltage sensitive channels or acid sensing channels and potentially better than 10% FBS cells for recording nAChR currents. The other channels found to be expressed in 1% FBS are also found in sensory neurons but are probably not expressed at high enough density to be experimentally useful.

Additional comments

The authors have made extensive improvements to the text and figures and have address most of the reviewer comments sufficiently.

---

## Round 0.3 · accepted · Accept

The authors have constructively addressed the final remaining comments on this manuscript.